# Regional Variety of Reduction in Retinal Thickness of Diabetic Macular Edema after Anti-VEGF Treatment

**DOI:** 10.3390/medicina58070933

**Published:** 2022-07-14

**Authors:** Yutaka Yamada, Yoshihiro Takamura, Takehiro Matsumura, Makoto Gozawa, Masakazu Morioka, Masaru Inatani

**Affiliations:** Department of Ophthalmology, Faculty of Medical Sciences, University of Fukui, Yoshida, Fukui 910-1193, Japan; yyutaka@u-fukui.ac.jp (Y.Y.); tmatsu@u-fukui.ac.jp (T.M.); makoto.gozawa@gmail.com (M.G.); mmorioka@u-fukui.ac.jp (M.M.); inatani@u-fukui.ac.jp (M.I.)

**Keywords:** diabetic macular edema, VEGF, vascular endothelial growth factor, microaneurysm

## Abstract

*Background and Objectives*: The presence of refractory cases resistant to anti-vascular endothelial growth factor (VEGF) therapy for diabetic macular edema (DME) is a problem in clinical practice. This study aimed to explore the less responsive area of optical coherence tomography (OCT) 3D map the characteristics of naïve DME cases after their first anti-VEGF. *Materials and Methods:* In 46 patients with DME who received an intravitreal injection of anti-VEGF agents, retinal thickness in 100 sections of the macular area was measured by 3D-mapping mode using OCT before and 1 month after injection. The density of the microaneurysm (MA) was calculated using merged images of the OCT map and fluorescein angiography. *Results*: One month after injection, the central retinal thickness significantly decreased (*p* < 0.0001). In severe edema (retinal thickness more than 500 µm), the area percentages with a reduction rate of the retinal thickness greater than 30% and less than 5% were 6.4 ± 6.6% and 10.1 ± 4.6%, respectively. The reduction rate of the retinal thickness varied from section to section. The mutual distance between the areas of maximum thickness before and after the injection averaged 1.22 ± 0.62 mm apart. The reduction rate of retinal thickness in the thickest region before injection was significantly higher (*p* = 0.02), and that in the thickest region after injection was lower (*p* = 0.001) than in the other regions. MA density in the residual edema was significantly higher than in the edema-absorbed area (*p* = 0.03). *Conclusion:* DME has areas that show low response to the reduction in retinal thickness with anti-VEGF therapy. A high density of MA may be associated with this pathogenesis.

## 1. Introduction

Diabetic macular edema (DME) is a major complication in patients with diabetes mellitus (DM) that can lead to vision loss [1,2]. DME occurs due to a disruption of the blood–retinal barrier, which increases the vascular permeability of capillaries and microaneurysms (MAs) [3]. DME can be differentiated into focal and diffuse types. MAs are associated with the pathogenesis of both types of DME [4]. The intraocular concentrations of vascular endothelial growth factor (VEGF) are higher in patients with DME than in those without DME [5]. Many randomized clinical trials have shown that anti-VEGF therapy improves visual acuity and macular swelling [6,7,8,9]. In Japan, two VEGF inhibitors, aflibercept and ranibizumab, have been approved for DME treatment by intravitreal injection.

Ophthalmologists currently recognize that the intravitreal injection of anti-VEGF drugs is the first-line treatment for DME [10,11]. However, multiple injections are required to maintain therapeutic efficacy. Moreover, anti-VEGF agents are expensive and thus impose a significant financial burden on patients [12,13]. Approximately 40% of cases are reported to be refractory to anti-VEGF therapy [14]. Several studies have examined the factors that predict a decreased response to anti-VEGF therapy. Optical coherence tomography (OCT) analysis has shown that hyperreflective foci in the cystoid space of the fovea predict a decreased short-term response to ranibizumab [15]. Previously, we reported that MA was densely distributed in residual edema after anti-VEGF injection [16]. Hirano et al. reported that more frequent ranibizumab injections for DME were associated with MA [17,18]. It is clinically important to identify the pathophysiology of inadequate response to anti-VEGF therapy to achieve adequate outcomes with fewer injections.

Until now, it was unclear whether the response to anti-VEGF therapy differed in the area of macular edema. OCT maps are useful for geographically assessing the edema-ameliorating effect of anti-VEGF therapy. In this tool, the grading of retinal thickness is depicted by color, and severe edema greater than approximately 500 µm is indicated by white color. However, this method is insufficient for analyzing detailed retinal thickness changes in each edema area. This study aimed to explore the characteristics of areas showing a poor response to intravitreal injections of anti-VEGF drugs for DME using a 3D mode. In this study, we utilized the 3D mode, in which the macular area was divided into 100 sections, which enabled us to measure the retinal thickness in each area. We used the 3D mode to determine whether there were geographic differences in improving retinal edema with anti-VEGF therapy. In addition, we explored the less responsive area of OCT 3D characteristics of naïve DME cases after their first anti-VEGF.

## 2. Materials and Methods

### 2.1. Study Design

This retrospective study adhered to the tenets of the Declaration of Helsinki, was approved by the Institutional Review Board of the University of Fukui and was registered with the University Hospital Medical Information Network Clinical Trials Registry of Japan (R000054484). Written informed consent was obtained from all patients after fully explaining the intent of the study to them. We included 46 eyes with DME that received anti-VEGF therapy between April 2017 and March 2019 at Fukui University Hospital. In bilateral cases, the first injected eye was used as the data.

### 2.2. Inclusion and Exclusion Criteria

The inclusion criteria were as follows: (1) age > 20 years; (2) a diagnosis of type 2 DM with center-involved DME; (3) the use of aflibercept or ranibizumab as anti-VEGF therapy for DME; and (4) macular thickening map image by OCT performed up to 1 month before anti-VEGF drug administration. The exclusion criteria were: (1) typical focal DME with ring-shaped hard exudate; (2) other retinal diseases, such as retinal vein occlusion or uveitis; (3) anti-VEGF therapy with focal/grid laser photocoagulation; (4) excessively severe medial opacity that precluded fundal evaluation (e.g., severe cataract, corneal opacity, or vitreous hemorrhage); and (5) the history of pan-retinal photocoagulation (PRP) and intraocular surgery for 6 months before and after the initial injection of the anti-VEGF agent.

### 2.3. Procedures

All patients underwent comprehensive ophthalmic examinations during their initial visits, including best-corrected visual acuity (BCVA) testing, slit-lamp biomicroscopy, intraocular pressure measurement using a Goldmann applanation tonometer (Luneau Technology Operations, Normandie, France) and dilated fundoscopy. Fluorescein angiography (FA) images were captured using Spectralis Heidelberg Retinal angiography (Heidelberg Engineering, Heidelberg, Germany); the OCT map, including the central retinal thickness (CRT) measurement, was captured using a Triton OCT (Topcon Medical Systems, Inc., Oakland, NJ, USA). All imaging tests were performed by experienced orthoptists blinded to the treatment status.

Intravitreal injections were administered in a standardized manner by a trained ophthalmologist (Y.Y.) using 0.4% oxybuprocaine hydrochloride (0.4% benoxyl ophthalmic solution, Santen Co. Ltd., Osaka, Japan) and 2% xylocaine as the anesthetic and povidone-iodine as an antiseptic. An eyelid speculum (HOYA CORPORATION, Tokyo, Japan) was used to stabilize eyelids. The injection concentration of aflibercept (Eylea, Bayer Yakuhin Ltd., Tokyo, Japan) was 2 mg/0.05 mL, while that of ranibizumab (Lucentis; Novartis Pharma K. K., Tokyo, Japan) was 0.5 mg/0.05 mL. Finally, an OCT examination was performed 1 month after the injection.

### 2.4. Retinal Thickness Measurement Using 3D OCT Mode

To examine the geographic pattern of retinal thickness, we used a 3D-map mode. The retinal thicknesses were displayed in descending order in white, red, yellow and green on the OCT map. Imaging with red and white indicated severe edema. The border between red and yellow was approximately 350 µm, and we defined more than 350 µm as an edematous area in this study. In addition, the border between white and red was approximately 500 µm, and we defined a severely edematous area of more than 500 µm. The macular area (6 mm × 6 mm square) centered on the fovea was divided into 100 (10 × 10) sections. The retinal thickness in each area was measured before and 1 month after the injection of the anti-VEGF agent. The area with the maximum retinal thickness before or after treatment was identified and the distance between them was measured. The reduction rate of retinal thickness was calculated as a percentage by dividing the retinal thickness 1 month after injection by the pretreatment retinal thickness.

### 2.5. Density of Microaneurysms Measurements by Merged Images

FA images were taken within 1 minute of the intravenous injection of fluorescein dye. Using Adobe Photoshop Elements (Adobe Systems Inc., San Jose, CA, USA), the FA images were merged with the OCT map images taken before or 1 month after the intravitreal injection of the anti-VEGF agent, as previously reported [19]. Residual edema was defined as the area shown in white (more than 500 µm in retinal thickness) on the OCT map after injection. The size of the edematous area was calculated as the percentage of a 6mm diameter circle centered on the fovea as 100%, and the density was calculated by dividing the number of MAs present in each area. The above steps were performed manually by three examiners (Y. Y., Y. T., and M. M.). Each examiner analyzed all enrolled patients, and the examiners masked the data counted by each examiner. The average values from the three examiners were used as the measured values of the patients.

### 2.6. Statistical Analysis

Statistical analyses were performed using the JMP software (SAS Institute Inc., Tokyo, Japan). Variables are expressed as mean ± standard deviation. Significant differences between different time points were analyzed using the Wilcoxon signed-rank test. Differences between groups were analyzed using the Mann–Whitney U test and were considered statistically significant at *p* < 0.05.

## 3. Results

We enrolled 46 patients who had received a single injection of an anti-VEGF agent. The baseline patient characteristics are summarized in Appendix A. Compared to baseline, CRT (from 515.27 ± 98.03 to 349.48 ± 85.02, *p* < 0.0001) and BCVA (from 0.52 ± 0.36 to 0.35 ± 0.34, *p* = 0.03) improved significantly at 1 month after treatment (Figure 1). In this study, 56.5% (26/46) of patients had a history of PRP. Between the patients with and without a history of PRP, there were no significant differences in the CRT (*p* = 0.207), BCVA (*p* = 0.285) and the rate of residual edema (*p* = 0.561) at 1 month after injection.

We divided the macular region into 100 sections, measured the retinal thickness in each section area in 3D map mode before and 1 month after injection and then calculated the reduction rate of retinal thickness. A representative case is shown in Figure 2A. Compared to pre-injection (Figure 2A(a)), the area of severe edema (white area in OCT map) became small at 1 month after the injection of the anti-VEGF agent (Figure 2A(c)). Sections within the white areas on the OCT map were grouped according to the degree of reduction in retinal thickness (Figure 2B).

These analyses were performed in parallel in all the cases. As shown in Figure 2C, areas showing larger reductions greater than 30% in retinal thickness accounted for 6.4 ± 6.6% of all edema, and areas showing a reduction of less than 5% accounted for 10.1 ± 4.6%, indicating that the rate of edema improvement varied from section to section.

The thickest areas before and after injection are marked with green and red rectangles, respectively, and the distances between them were measured (Figure 3A).

The mean length for all cases was 1.22 ± 0.62 mm (vitrectomized eyes: 1.19 ± 0.23 mm, non-vitrectomized eyes: 1.22 ± 0.69 mm). There was a significant positive correlation between the distance between the thickest areas before and after injection and the size of the macular edema (retinal thickness greater than 500 µm) before injection (*p* = 0.002, Y = 0.03X + 0.35, R^2^ = 0.20). The rate of retinal thickness reduction in the thickest region before injection was significantly greater than that in the other regions (*p* = 0.02), and in the thickest region after injection, it was significantly lower (*p* = 0.001).

To assess the density of MA in the residual edema, we merged the FA images and OCT maps taken before or 1 month after injection (Figure 4A). On the FA image, MAs were marked, and the density of MA in the residual and absorbed areas was calculated. One month after injection, 21 patients (45.7%) had residual edema with 500 μm or greater retinal thickness (aflibercept 20/42, ranibizumab: 1/4). Four cases with a history of vitrectomy showed no residual edema after the initial injection. The reduction rate of retinal thickness in the residual edema group was significantly lower than that in the edema-absorbed area (Figure 4B). MA density in the residual edema was significantly higher than in the edema-absorbed area (*p* = 0.03) (Figure 4C).

Additional injections were given to 21 patients with residual edema after the first injection. At 1 month after the second injection, 38.1% (8/21) of the patients still had residual edematous areas. All (62.5%, 5/8) or some (37.5%, 3/8) of the residual edematous area after the second injection was contained within the area of residual edema after the first injection.

We compared the changes in CRT and BCVA in the patient groups with and without the residual edema after the initial injection (Figure 5). Although we found no significant difference in BCVA between the groups, the CRT in the eyes without the residual edema was significantly smaller than that with the residual edema at 1 (*p* = 0.008) and 3 months (*p* = 0.04). The number of injections of the anti-VEGF agent in the eyes with the residual edema (2.37 ± 1.01) was significantly greater (*p* = 0.0002) than those in the eyes without the residual edema (1.46 ± 0.53). A significant correlation was found between the width of the residual edema and the number of injections during 6 months (*p* = 0.01).

## 4. Discussion

In this study, we found that the reduction rate of retinal thickness after anti-VEGF therapy was distributed to various degrees in regions of DME. OCT maps were utilized to determine the geographic pattern of retinal thickness in the macular region, with areas of severe edema (approximately 500 µm or more) displayed in white. The change from white to red, yellow or green on the OCT map after the injection of anti-VEGF agents indicates an improvement in retinal thickness; however, color information alone could not provide a detailed analysis of the reduction rate of retinal thickness. Therefore, we subdivided the edematous region into 100 areas using the 3D macular mode and examined the reduction rate of retinal thickness in each area after the anti-VEGF injection. Based on our data, the areas that showed a 10–20% reduction in retinal thickness accounted for approximately 40% of the total edema areas.

Meanwhile, only about 6.4% of the edema areas showed a dramatic reduction of 30% or more in the retinal thickness. The areas where the reduction in retinal thickness was less than 5%, indicating refractoriness to anti-VEGF therapy, accounted for approximately 10% of the area of edema. These results suggest that the edema-ameliorating effect of anti-VEGF therapy varies by site and that there are areas that are less responsive to the drug. Our data showed that the distance of the maximum areas before and after injection was longer, with a larger area of the edema region. This result implies that the thickest region before injection did not remain the thickest region after injection. These results further strengthen the credibility of the finding that the effect of anti-VEGF treatment on retinal thickness varies by region. The average reduction rate of retinal thickness was significantly lower in the thickest areas after injection. It is considered that this low responsiveness to anti-VEGF drugs leads to residual edema. The edematous area that remained after the second injection was contained within the residual area after the first injection. This result suggests that the retinal areas resistant to anti-VEGF treatment keep a similar localization of the residual edema after additional injection.

It is clinically important to elucidate the factors that make anti-VEGF therapy less effective in ameliorating residual edema. Similarly to the results of this study, we have recently shown that MA density is higher in areas of residual edema after anti-VEGF treatment for DME [16]. DME caused by MA can be classified as focal DME; it is distinguished from diffuse DME caused by extensive hyperpermeability [1]. It has been reported that the therapeutic effect of anti-VEGF drugs tends to be less effective in focal DME than in diffuse DME [20]. Hirano et al. showed that the number of ranibizumab injections required to achieve a stable improvement in edema was significantly higher when MA was present in the neighboring foveal avascular zone (FAZ) in the center of DME [17,21]. In addition, Lee et al. reported that OCT angiographic analysis revealed that small vascular flow density in the deep capillary plexus, large FAZ area, and a large number of adjacent MAs are characteristics of cases with poor response to anti-VEGF therapy [22]. These findings support the possibility that MA is a factor contributing to resistance to anti-VEGF therapy.

If a high MA density is a risk factor for residual edema after anti-VEGF therapy, what treatment strategies should ophthalmic clinicians develop? We reported that with repeated additional injections of anti-VEGF drugs for the residual edema with high dense MA, the resolution of edema could be achieved in approximately 90% of cases [16]. Our data in the current study also showed that more frequent injections were performed in the eyes with residual edema. Moreover, a significant correlation was found between the width of the residual edema and the number of injections for 6 months, indicating that more injections are required to treat the wide area of the residual edema. Despite the greater number of injections, the reduction in CRT was smaller in the group with residual edema than in the group without residual edema, while the visual outcome was comparable. Although an increased number of injections is required, frequent injections can improve visual prognosis, even in cases refractory to anti-VEGF therapy. Direct laser photocoagulation targeting the MAs may also be effective as an alternative treatment option. Although it is difficult to identify the location of MAs accurately, we have shown that merging FA images depicting MAs with OCT maps and fundus photographs improve the outcomes of focal laser therapy for focal DME [19]. In recent years, navigation laser systems have been developed, making it possible to apply precise laser treatment to MAs automatically. [23,24]

The limitation of this study is the small sample size. Factors such as the differences in drugs and the history of vitrectomy may influence anti-VEGF treatment. Further analysis with a larger number of patients is necessary.

## 5. Conclusions

In conclusion, this study has revealed that the edema ameliorating effect of a single injection of anti-VEGF drugs on DME varied from site to site. A high density of MA appears to be involved in this pathogenesis. Future investigation of what additional treatment should be administered to these refractory areas is important for more effective anti-VEGF therapy.

## Figures and Tables

**Figure 1 medicina-58-00933-f001:**
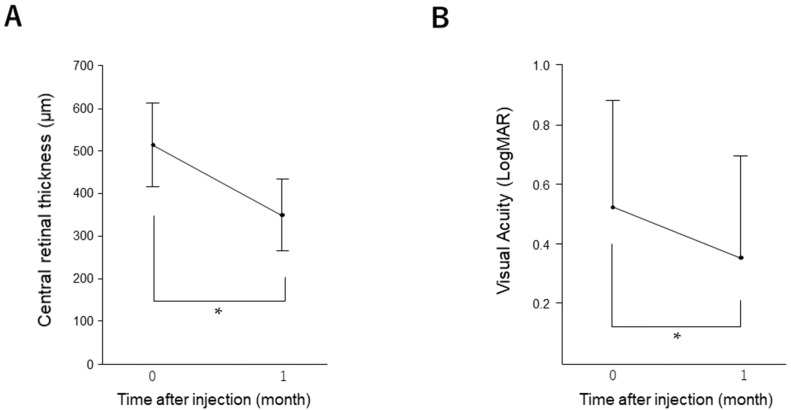
Changes in CRT and BCVA after injection of anti-VEGF agents. CRT (**A**) and BCVA (**B**) were measured 0 and 1 month after a single injection. BCVA is expressed as logMAR. Data are presented as mean ± standard deviation (SD). * *p* < 0.05 (versus baseline). CRT, central retinal thickness; BCVA, best-corrected visual acuity; VEGF, vascular endothelial growth factor; SD, standard deviation.

**Figure 2 medicina-58-00933-f002:**
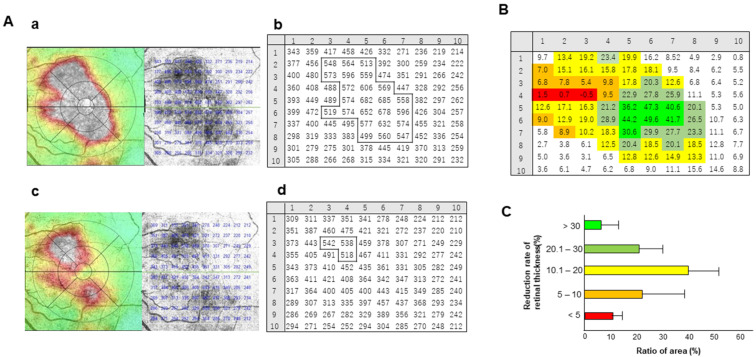
Distribution pattern of the reduction rate of retinal thickness. (**A**) Representative case of OCT map image and 3D-map of retinal thickness before (**a**,**b**) and 1 month after injection (**c**,**d**) of anti-VEGF agents. The macular area (6 mm × 6 mm square) was divided into 100 sections (10 × 10 squares), and the retinal thickness in each section was measured (**b**,**d**). Areas greater than 500 µm in retinal thickness were outlined. OCT, optical coherence tomography. (**B**) Distribution pattern of the retinal thickness reduction rate after anti-VEGF treatment in a representative case. The light green, dark green, yellow, orange, and red colors indicate the areas where the reduction ratio was more than 30%, 20.1–30%, 10.1–20%, 5.1–10% and less than 5%, respectively. (**C**) Proportion of areas showing different retinal thickness reduction rates in all cases.

**Figure 3 medicina-58-00933-f003:**
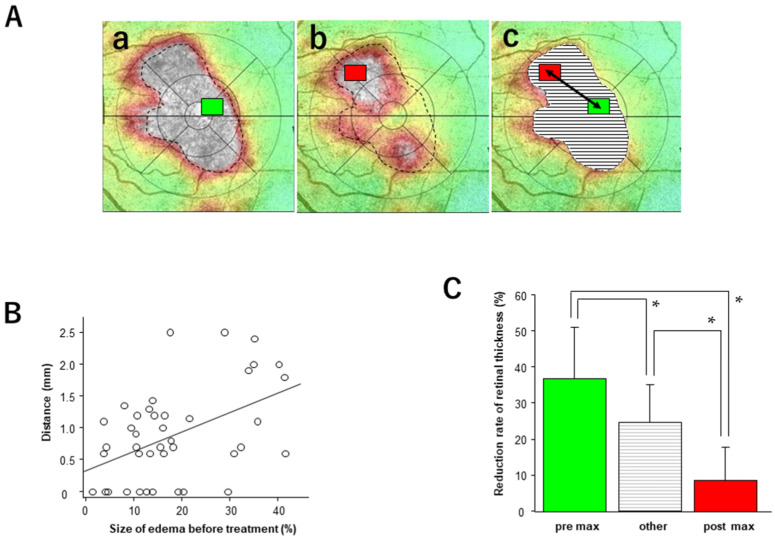
Analysis of the areas with maximum retinal thickness before and after injection. (**A**) OCT map of a representative case (**a**–**c**). The broken line indicates an outline of the severely edematous area. Green and red rectangles indicate the maximum area before (**a**) and 1 month after injection (**b**). The distance between the center of the green and red rectangles (arrow) is measured (**c**). The other areas are indicated by horizontal borders. (**B**) Linear correlation between the distance of the maximum thickness area and the size of the severe edema before injection. Significant correlations were observed (*p* = 0.002, R^2^ = 0.20). (**C**) Comparison of the reduction rate of retinal thickness among the areas with maximum thickness before and after injection and the other areas. The color and texture link is shown in (**A**). Data represent mean ± standard deviation (SD). * *p* < 0.05.

**Figure 4 medicina-58-00933-f004:**
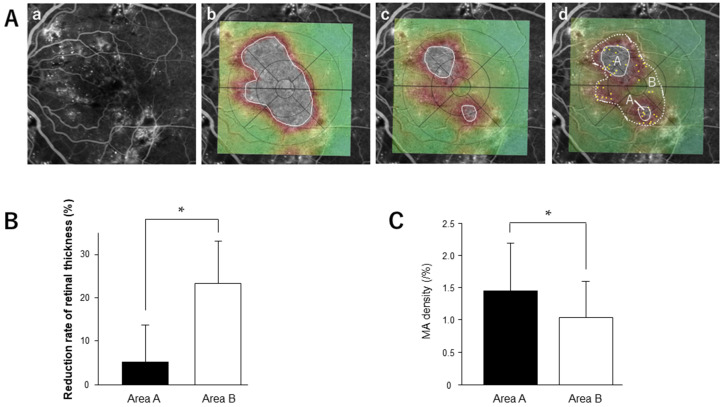
MA density and reduction rate of retinal thickness in the residual edema after anti-VEGF treatment. (**A**) Representative merged image. Fluorescein angiography images taken before injection (**a**) were merged with the OCT maps taken before (**b**) and 1 month after injection (**c**). The severely edematous area (>500 µm) is outlined by a white line. The MA is marked in the merged image (**d**). A: Residual edema area; B: edema absorption area. (**B**) Comparison of the reduction rate of retinal thickness between the residual edema and the edema-absorbed area. Area A: Residual edema; Area B: edema absorption area. Data represent mean ± standard deviation (SD). * *p* < 0.05. (**C**) Comparison of MA density between the residual edema and edema-absorbed area. Area A: Residual edema; Area B: edema absorption area. Data represent mean ± standard deviation (SD). * *p* < 0.05. MA, microaneurysms.

**Figure 5 medicina-58-00933-f005:**
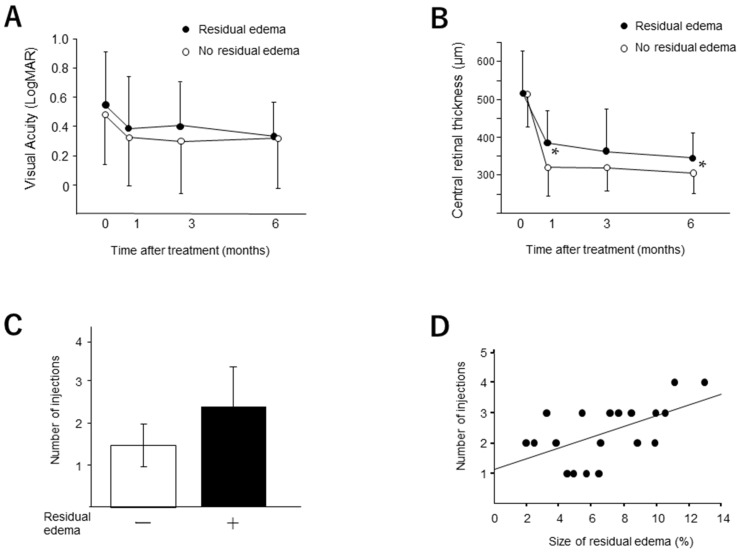
The involvement of the residual edema for anti-VEGF therapy. The changes of visual acuity (**A**) and central retinal thickness (**B**) in the eyes with and without the residual edema were measured at 0, 3 and 6 months. Visual acuity is expressed as logMAR. Data are presented as mean ± standard deviation (SD). * *p* < 0.05 (versus groups). (**C**) The number of injections of anti-VEGF agent during 6 months. (**D**) The correlation between the number of injections and the size of residual edema after initial injection. Significant correlations were observed (*p* = 0.0002, R^2^ =0.32).

## Data Availability

The datasets generated during and/or analyzed during the current study are available from the corresponding author on reasonable request.

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
