# Peer review of "Regional Variety of Reduction in Retinal Thickness of Diabetic Macular Edema after Anti-VEGF Treatment"

_medicina, 2022, doi:10.3390/medicina58070933_

Round 1
Reviewer 1 Report
The study of Yamada et al shows that, in a population of type 2 diabetic patients with diabetic macular edema (DME) treated with anti-VEGF agents, the reduction of thickness was not homogeneous along the entire retina with areas more resistant and areas more sensitive to the anti-VEGF treatment. Microaneurisms (MA)were more frequent in the resistant areas and the Authors suggest that this could have an impact in the pathogenesis of DME.
Comments to the manuscript are:
1) This study shows that there is a correlation between retinal areas resistant to anti-VEGF treatment and presence of MA in diabetic patients affected by DME.
At the moment of the study, however, more than half of the patients were however already treated with pan retinal photocoagulation (PRP). This represents a major confounding factor as PRP could have altered the distribution of MA inside the retina.
2) Is there a correlation between PRP-treated areas and areas resistant to anti-VEGF treatment in the retina of these patients?
3) According to the Methods section these patients received a number of injections of anti-VEGF agents, but only the effect of the first injection was considered for this study.
It would be very interesting to evaluate, in a longitudinal way, whether the retinal areas resistant to anti-VEGF treatment remain the same along time or if there is a “movement” of these regions.
4) Is there a correlation between number of retinal areas resistant to anti-VEGF treatment and prognosis of DME in the same patient?
Reviewer 2 Report
The study investigated the effect of anti-VEGF on 3D OCT map of macula. Generally, the draft is fluent is good and the strategy for presentation is organized. The conclusion is adequate.
1. The study aimed to study “the characteristics of areas showing a poor response to intravitreal injections of anti-VEGF drugs for DME”. However, in fact the authors studied the less responsive area of OCT 3D map characteristics of naïve DME cases after their first anti-VEGF. The wordings are misleading.
2. Since either afibercept or lucentis was used for intravitreal management of DME, was there a difference in MA regression observed? Or a difference in the edema area resolution observed?
3. Material and methods. Line 72. What do the authors mean by consecutive eyes?
4. Material and methods. Patients who had vitrectomy in the past will change the constitution of their vitreous and may not be suitable to include in the analysis.
Round 2
Reviewer 1 Report
The manuscript has been modified according to the suggestions